# Improvement of Diet after an Early Nutritional Intervention in Pediatric Oncology

**DOI:** 10.3390/children10040667

**Published:** 2023-03-31

**Authors:** Mélanie Napartuk, Véronique Bélanger, Isabelle Bouchard, Caroline Meloche, Daniel Curnier, Serge Sultan, Caroline Laverdière, Daniel Sinnett, Valérie Marcil

**Affiliations:** 1Research Center of the CHU Sainte-Justine, Montreal, QC H3T 1C5, Canada; 2Department of Nutrition, Faculty of Medicine, Université de Montréal, Montreal, QC H3T 1A8, Canada; 3School of Kinesiology and Physical Activity Sciences, Université de Montréal, Montreal, QC H3G 1Y5, Canada; 4Department of Psychology, Université de Montréal, Montreal, QC H3C 3J7, Canada; 5Department of Pediatrics, Faculty of Medicine, Université de Montréal, Montreal, QC H3T 1C5, Canada

**Keywords:** child, adolescent, cancer, nutrition, intervention, diet quality score, cardiometabolic health

## Abstract

Pediatric cancer survivors may experience cardiometabolic sequelae over the course of their lives as a result of the treatments they have received. While nutrition consists of an actionable target for cardiometabolic health, few nutritional interventions have been documented in this population. This study assessed the changes in diet during a one-year nutritional intervention for children and adolescents undergoing cancer treatments and the participants’ anthropometric and cardiometabolic profiles. A total of 36 children and adolescents (mean age: 7.9 years, 52.8% male) newly diagnosed with cancer (50% leukemia) and their parents underwent a one-year individualized nutrition intervention. The mean number of follow-up visits with the dietitian during the intervention was 4.72 ± 1.06. Between the initial and one-year assessments, there was an improvement in diet quality reflected by the Diet Quality Index (5.22 ± 9.95, *p* = 0.003). Similarly, the proportion of participants with moderate and good adherence (vs. low adherence) to the Healthy Diet Index score almost tripled after one year of intervention (14% vs. 39%, *p* = 0.012). In parallel, there was an increase in the mean z-scores for weight (0.29 ± 0.70, *p* = 0.019) and BMI (0.50 ± 0.88, *p* = 0.002), and in the mean levels of HDL-C (0.27 ± 0.37 mmol/L, *p* = 0.002) and 25-hydroxy vitamin D (14.5 ± 28.1 mmol/L, *p* = 0.03). Overall, this study supports that a one-year nutritional intervention deployed early after a pediatric cancer diagnosis is associated with an improvement in the diets of children and adolescents.

## 1. Introduction

In North America, the evolution of antineoplastic therapies, refinement of resistance/relapse risk stratification, and optimization of supportive care have widely contributed to improving the survival rates of cancer in children and adolescents [1,2]. Over the last decades, the five-year survival rate for childhood cancer has steadily increased, reaching over 80% in developed countries [2,3]. Nevertheless, treatments cause significant late adverse effects in the growing population of childhood cancer survivors (CCSs) [4]. While antineoplastic agents target malignant tissues with high proliferation rates, they are not specific. Hence, healthy cells can also be damaged, causing unwanted acute and long-term side effects [5,6]. In the long-term, CCSs have a 40–70% higher risk of late mortality or morbidities than the general population [7,8,9,10] including possible secondary cancer, osteoporosis, obesity, cardiovascular, neurological, and fertility complications [11,12,13].

Cardiovascular complications are an area of concern for adult survivors of childhood cancer as they are a significant cause of early mortality [14,15] and the principal non-neoplastic cause of death [16]. In children, the prevalence of overweight increases during cancer treatment from 8% at diagnosis to 13% at the end of treatment, regardless of diagnosis [17]. This issue is critical since a retrospective review on acute myeloid leukemia highlighted that patients aged 1 to 20 years, who were overweight, were more likely to experience treatment-related mortality than those who were middleweight [18]. The use of glucocorticoids, radiation therapy, and stem cell transplantation are associated with a higher risk of insulin resistance in survivors of acute lymphocytic leukemia (ALL) and non-Hodgkin lymphoma [19,20,21]. According to studies conducted in the induction and maintenance phases of treatment, the proportion of ALL pediatric patients who develop insulin resistance can be as high as 50% [22,23]. Lipid abnormalities (high triglycerides, low-density lipoprotein cholesterol (LDL-C) or low high-density lipoprotein cholesterol (HDL-C)) have been observed in up to 50% of childhood ALL survivors [24]. In addition, studies have reported that between 15.3% and 46.4% of CCS become hypertensive or prehypertensive during their lifetime [25,26,27,28].

The exact mechanisms contributing to the development of cardiometabolic complications during and after pediatric oncologic treatment are unknown. In the general population, it is well-documented that dietary habits significantly impact cardiovascular and metabolic health [29,30,31]. Unfortunately, evidence shows that children with cancer consume many poor-quality foods, exacerbated when corticosteroids are used in treatment modalities [32,33,34,35]. The findings of an exploratory study [33] demonstrated that energy intake in children undergoing oncology treatment was adequate. Still, the food quality was generally poor, as 94% did not meet the recommendations for vegetables, 77% for fruits, and 75% for dairy products. Dietary hyperselectivity is often encountered [36], combined with a preference for carbohydrate-rich foods such as bread, pasta, rice, and potato dishes [33,37]. This may have an inordinate impact on the youth’s health since nutritional status in pediatric oncology influences prognosis [38,39] and modulates side effects during [40,41] and after treatment [42]. Some authors have emphasized the difficulty of reversing unhealthy eating behaviors acquired during treatment [43], and several suggested implementing screening and intervention protocols at diagnosis and throughout treatment [44,45]. Despite these recommendations and the potential impact of diet during and after pediatric cancer, few nutrition interventions have been deployed with this population, and only one initiated early after diagnosis has been documented [46]. We have previously described the feasibility of an early nutrition intervention for children and adolescents undergoing cancer treatment [47]. Here, we assessed the magnitudes of changes in diet during an early nutrition intervention in pediatric oncology and monitored in parallel the variation in the participants’ anthropometric and cardiometabolic profiles.

## 2. Materials and Methods

### 2.1. Study Design

This research is part of the VIE Study (Valorization, Implication, Education) that consists of an integrated intervention program for children and adolescents newly diagnosed with cancer. The multicomponent intervention, implemented early after diagnosis, aims to provide nutrition, physical activity, and psychological support to patients and their families to prevent short- and long-term health complications [47,48]. Recruitment was carried out between February 2018 and December 2019 at the Charles-Bruneau Oncology Center of the CHU Sainte-Justine (CHUSJ) (Montreal, Canada). The VIE study was approved by the Ethics Review Board of the CHUSJ (#2017-1413) and conducted in accordance with the Declaration of Helsinki.

### 2.2. Participants

As described elsewhere [47], 147 newly diagnosed pediatric cancer patients who were going to be treated at the CHUSJ were screened by the clinical coordinator during the study recruitment period. Potential participants were: (1) aged under 21 years; (2) being treated by chemotherapy and/or radiotherapy. The attending oncologist and the health care team of each potential patient were contacted to obtain (3) approval for participation in the VIE study. Eligible participants (*n* = 92) were approached by the clinical coordinator for study presentation if time since diagnosis was between 4 and 12 weeks and if the three inclusion criteria were met. Recruited participants (*n* = 62) provided written informed consent (from a parent or legal guardian). At any time during the study, the attending oncologist had the possibility to withdraw the participant if their health condition no longer allowed them to participate in the intervention.

### 2.3. Nutrition Intervention

The nutritional intervention of the VIE project was offered in addition to standard clinical nutrition care. The details of the intervention and its feasibility have been published elsewhere [47]. Briefly, the intervention consisted of an initial assessment, follow-up visits planned every two months for one year, and a one-year assessment. The first meeting with a research registered dietitian (RD) aimed to establish contact with the participant and his family and create a bond of trust. The initial and one-year assessments included clinical, anthropometric, biochemical, and nutritional data to describe changes in metabolic health and nutritional status. On follow-up visits, only clinical, anthropometric, and nutritional data were collected (Figure 1). Throughout the intervention, RDs monitored changes in weight, blood pressure, and diet and provided individualized counselling addressing the patient’s challenges and concerns. Visits were conducted in the hospital room if the child was hospitalized or at the outpatient oncology clinic. From March 2020 until the end of the project, follow-ups were held by phone due to COVID-19. The intervention approach was based on motivational interviewing, encouraging patients to improve their diet, and parents to accompany their children in their new challenges. The approach also aimed at alleviating the food and weight-related guilt of the patients and parents. As previously described [47], after one year of intervention, the level of engagement was determined subjectively by the RDs based on the patient’s motivation to attend the meetings, discuss food, and adapt their diet. Level of engagement was listed as: low—the participant showed minimal involvement, multiple refusals, and/or avoidance of meetings; moderate—the participant was passively involved in visits and/or if multiple appointments were required to complete a follow-up; high—the participant was actively engaged during visits and ease of scheduling/conducting follow-up visits. The number of follow-up visits was separated into two categories: ≤4 and ≥5 visits.

### 2.4. Dietary Assessment Data

Dietary data were collected at the initial assessment and each follow-up visit using a 24-h dietary recall. A 3-day food record (2 weekdays and one weekend day) was provided to the patient and family at the initial and one-year assessments. Nutritional data were analyzed using the Nutrific^®^ software developed by the Department of Food Science and Nutrition of Université Laval based on the 2010 Canadian Nutrient File. Total daily energy (kcal) was computed, and nutrient intakes were adjusted for total energy (per 1000 kcal). Dietary intakes of macronutrients, vitamin D, sodium, calcium, omega-3 polyunsaturated fatty acids, and fibers were computed and analyzed. Two scores of diet quality were calculated: the Diet Quality Index (DQI) [49] and the Healthy Diet Index (HDI) [50]. While the DQI score is a continuous score of 100 points, the HDI score is summarized in three categories of low, medium, or good depending on the level of adherence to the nutritional recommendations. Finally, the percentage of daily calories from ultra-processed foods (UPF) on the total daily calories was calculated based on the NOVA classification [51].

### 2.5. Anthropometric Evaluation and Blood Pressure

Anthropometric measurements and blood pressure were taken at the initial and one-year assessments. Body weight was measured using a calibrated electronic scale and recorded to the nearest 0.1 kg. Height was measured with a stadiometer to the nearest 0.1 cm. Body mass index (BMI) was calculated from the weight and height (kg/m^2^). Waist circumference (WC), mid-upper arm circumference (MUAC), triceps skinfold thickness (TSFT), and subscapular skinfold thickness (SSFT) were measured by trained dietitians according to the validated National Health and Nutrition Examination Survey (NHANES) protocols [52,53,54,55]. WC and MUAC were measured with a non-stretchable measuring tape to the nearest 0.1 cm at the level of the iliac crest in the mid-axillary line and midway between the acromion and olecranon processes of the ulna, respectively [52,53,55]. TSFT was measured at mid-arm and SSFT was recorded below the scapula at a 45 angle of the spine [52,54]. The mean of two consecutive measures to the nearest 0.2 mm with a Harpenden skinfold caliper was used to measure the triceps skinfold thickness (TSFT) and subscapular skinfold thickness (SSFT) if the platelet cell count was high enough to avoid bruising risks. To avoid inter-personal variation, all anthropometric measures of each participant were conducted by the same dietitian. Blood pressure (systolic and diastolic) measurements were collected on the same day from the medical chart. For each measure, the z-scores for age and sex were calculated using the Microsoft^®^ Office Excel^®^ tool developed by the British Columbia Children’s Hospital and the Canadian Pediatric Endocrine Group (version 2020) based on the Growth Charts for Canada [56]. This tool is valid for z-scores between −3 and 3. Therefore extreme z-scores were assessed separately with the World Health Organization references using a method described elsewhere [57].

### 2.6. Biochemical Assessment

Biochemical data were captured at the initial and one-year assessments. Blood sampling was coordinated by the care team and performed by the nurses. Using non-fasting serum and an Architect^®^-Ci8200 analyzer (Abbott, Chicago, IL, USA); glucose, total cholesterol (TC), triglycerides, and HDL-C were measured by photometry; insulin by chemiluminescent immunoassay; and high-sensitivity C reactive protein (CRP) by immunoturbidimetry. The Friedewald equation was applied to calculate the LDL-C. By subtracting HDL-C from TC, the non-HDL cholesterol value was calculated. HbA1c was measured using ion-exchange high-performance liquid chromatography. Liquid chromatography-tandem mass spectrometry was used to measure the serum 25-hydroxy vitamin D [25(OH)D].

### 2.7. Statistical Analyses

For descriptive statistics, continuous variables are presented as the mean ± standard deviation (SD) and minimum–maximum (min–max), and categorical variables as frequency and rates (*n*, %). Given that this was a feasibility study, the primary purpose of the analysis was to document the effect sizes rather than to validate the hypothesis tests. The paired t-test and Wilcoxon test were used to compare the importance of the differences between assessments for all continuous variables: DQI, HDI, NOVA (%), energy (kcal/d), total fat (g/1000 kcal/d), protein (g/1000 kcal/d), carbohydrate (g/1000 kcal/d), omega-3 fatty acids (g/1000 kcal/d), fiber (g/1000 kcal/d), weight z-score, height z-score, BMI z-score, waist circumference z-score, MUAC z-score, TSFT z-score, SSFT z-score, z-score of systolic and diastolic blood pressure, HbA1c (%), 25(OH)D (mmol/L), CRP (mmol/L), TC (mmol/L), HDL-C (mmol/L), LDL-C (mmol/L), non HDL-C (mmol/L), and triglycerides (mmol/L). The McNemar test was used to assess the magnitudes of changes in the diet determined by the HDI score categories (low adherence vs. medium and good adherence). In addition, the participants were stratified according to their level of participation (≤4 vs. ≥5 follow-up visits), engagement in the intervention (low and moderate vs. high), and sex (female vs. male). The importance of changes in diet (DQI, HDI, NOVA, nutrients) were compared between groups using unpaired t-tests. The use of parametric tests was favored to facilitate data interpretation, but non-parametric tests were also conducted. In the case of disparities between the conclusions from both tests, the one from the non-parametric test was presented.

## 3. Results

### 3.1. Description of the Cohort

The overall data collection process for the nutritional intervention is detailed in Figure 2. A total of 62 participants were enrolled in the VIE study, of which 60 were met for the initial dietary assessment. Six participants (9.7%) withdrew from the study because their medical condition worsened, and 9 (14.5%) dropped out for various reasons (no interest in pursuing, no interest during COVID, lost during follow-up). Therefore, 45 participants were eligible for the 6-month and 1-year evaluations. Dietary data were successfully collected at both the initial and 1-year assessments for 36 participants (58%) and the initial, 6-month, and 1-year assessments for 34 participants (55%).

The sociodemographic characteristics of the 36 participants are detailed in Table 1. There were slightly more males (52.8%) than females (47.2%), and the mean age at initial assessment was 7.9 years. Based on their age at initial assessment, 2/3 were children (<10 years old) and 12 were adolescents (≥10 years old). Half of the cohort had a diagnosis of ALL and a quarter of lymphoma. On average, participants had 4.7 visits with the research RD during the 1-year intervention.

### 3.2. Evolution of Diet Quality Scores and Dietary Intakes during the Intervention

The differences in the diet quality scores, percentage calories from UPF, and intake in macro- and micronutrients between the initial assessment, after 6 months, and one year of intervention are shown in Table 2. The DQI score improved after one year compared to the initial (difference of 5.22 ± 9.95; *p* = 0.003) and 6-month evaluations, although the latter did not reach statistical significance (difference of 4.75 ± 12.98; *p* = 0.093). Similarly, the proportion of participants with moderate and strong adherence (vs. low adherence) to the HDI score was greater after 6 months (*n* = 12, 35.3%) and one year of intervention (*n* = 14, 38.9%) than at the initial assessment (*n* = 5, 13.9%). No changes in the percentage of calories from UPF were noted during the intervention. The mean nutrient intake did not differ from the initial assessment after one year, except for an increase in the total caloric intake (333 ± 1030; *p* = 0.033), which was already higher at 6 months of intervention (234 ± 622, *p* = 0.035). While the protein intake tended to be higher at 6 months compared to the initial assessment (difference of 5.51 ± 14.90, *p* = 0.072), all other macronutrient intakes remained stable at the 6-month and one-year evaluations. No difference was found in the mean calcium, sodium, and vitamin D intake at the three time points (data not shown). When comparing the nutritional data according to the number of RD visits (≤4 vs. ≥5) or commitment level (low and moderate vs. high), we found no difference between groups (data not shown).

Figure 3 shows the mean diet score evolution stratified by sex. After one year of intervention and compared to the initial assessment, there was an improvement in the DQI score in both boys and girls. The difference between the initial and one-year assessments was also detectable with the HDI score in girls only. While in girls, the improvement in the DQI score was only observed at the one-year assessment, in boys, there was a clear tendency toward an improvement at 6 months, although it did not reach statistical significance.

### 3.3. Evolution of Clinical and Anthropometrical Characteristics during the Intervention

The comparison of the clinical and anthropometric parameters between the two timepoints (initial and one year) is presented in Table 3. The mean z-scores of weight and BMI increased after one year of intervention, while the mean height z-scores decreased. No other clinical and anthropometric outcomes were changed. In addition, no difference in blood pressure was observed. When the data were stratified according to sex (Appendix A), the differences in the weight and BMI z-scores reached statistical significance only in the boys (the trends were similar in girls). Finally, we observed a slight decrease in the height z-score in girls (the trends were similar in boys).

### 3.4. Biochemical Assessment at Initial Evaluation and after One Year of Intervention

The comparison of biochemical data (means) between the initial evaluation and after one year is presented in Table 4. After one year of follow-up, we found an increase in the mean levels of HDL-C (difference of 0.27 ± 0.37, *p* = 0.002) and 25-hydroxy vitamin D (difference of 14.52 ± 28.08, *p* = 0.03). The mean TG levels were lower after one year in participants aged less than 9 years old (difference of −0.43 ± 0.81; *p* = 0.02). We further stratified the data by sex (Appendix A) and found that there was no statistically significant variation in boys, whereas, in girls, the mean HbA1c decreased by −0.60 ± 0.57 (*p* = 0.014) and the mean HDL-C increased by 0.32 ± 0.37 (*p* = 0.017).

## 4. Discussion

In this study, we observed an overall improvement in the diet quality after a one-year nutritional intervention implemented early after pediatric cancer diagnosis. To our knowledge, this is the first study reporting an improvement in diet quality using dietary scores in children and adolescents undergoing a nutritional intervention during cancer treatment. A total of 36 children and adolescents newly diagnosed with cancer and their family underwent a one-year individualized nutrition intervention aiming at preventing short- and long-term metabolic health complications. The two scores used to assess diet quality were improved after the intervention. The mean DQI score improved by 5.22 ± 9.95 points between the initial and one-year assessments. Likewise, compared to low adherence, the proportion of patients with moderate and good adherence to the HDI score increased from 14% to 39% between the two time points. In parallel, we found an increase in the mean z-scores for weight (0.29 ± 0.70) and BMI (0.50 ± 0.88), and in the mean levels of HDL-C levels (0.27 ± 0.37) and 25-hydroxy vitamin D (14.52 ± 28.08 mmol/L).

Other nutritional interventions in pediatric oncology have reported improvements in specific aspects of diet over time. Among them, in a multicenter pilot study implementing a nutrition intervention program shortly after a pediatric cancer diagnosis, the authors found an increase in the mean intakes of proteins and vegetables and a decrease in added sugar consumption when the final and initial assessments were compared [46]. Similar to our results, variations in caloric intakes were observed after the induction phase but did not persist over time. These results are comparable to those reported by Zhang et al. [58], who piloted the 12 week web-based program HEAL. From the beginning to the end of their intervention, the mean intake of protein was increased, while the added sugar mean intake was found to be lower. However, no difference was found in milk, calcium, and potato consumption, nor in energy and most nutrients (total fat, carbohydrates, omega-3, fiber, sodium, calcium, and vitamin D). These results are similar to ours given that little changes in the different nutrients were observed over time. However, in contrast, no correction for total caloric intake was performed. This can affect the overall results since the nutrient intake of children and adolescents naturally increases over time, in parallel with their growth. Finally, unlike us, changes in the Healthy Eating Index-2015 score were not statistically significant [58].

In our study, the evolution of diet quality scores was found to be different between boys and girls. Although, the overall improvement in DQI was visible in both groups after one year of nutritional intervention, in boys, the quality of the diet tended to improve after 6 months, whereas in girls, the changes were mainly detected during the second half of the intervention (between 6 and 12 months). Conversely, in a cohort of 132 adolescents in remission from cancer, a one-year intervention aiming at reducing health risk behaviors had no impact on the dietary practices of girls and had a rather negative effect on boys [59]. Since the female sex is known to be a risk factor for cancer treatment-related cardiotoxicity [14,15], our results support the importance of implementing long-term nutritional interventions to maximize the chances of positive outcomes in this subgroup. In addition, stratification according to the participants’ level of engagement or the number of follow-up visits attended did not reveal differences. We would have expected that patients who had more follow-up meetings or demonstrated a greater commitment to the intervention would have had a more substantial improvement in their diet score. The limited sample size may have not allowed us to capture such findings.

Other studies have observed similar changes in weight and BMI during a nutrition intervention [60,61]. One intervention implemented with children and adolescents during the maintenance phase of ALL treatments reported a lesser increase in weight and BMI in the intervention group than in the control group, representing a positive outcome [60]. Since it is known that high BMI at diagnosis is associated with obesity in survivorship [62], one study targeted only children and adolescents that were overweight at diagnosis [61]. The authors found that adolescents over 14 years old in the intervention group gained less weight than those in the control group. Conversely, other nutrition intervention studies did not observe differences in the anthropometric measures of the children and adolescents after a lifestyle intervention [34,46,58,63,64,65]. When stratifying for sex, we found differences in the evolution of anthropometric data between boys and girls. While weight and BMI increased in the boys, we detected a slowing of growth in girls. The latter may contribute to the magnitude of the BMI increase. Skinfolds and brachial circumference may be more accurate to assess changes in lean body mass and bypass the weight effect of cancer masses, fluid retention, and edema [66,67]. However, in practice, obtaining these measurements remains challenging in this population. In our cohort, we reported a substantial number of missing data, as explained elsewhere [47]. Briefly, measurements were not completed to respect the patient’s medical condition, cooperation, and/or privacy. Notwithstanding, we distinguished no difference over time in the waist or brachial circumference or skinfolds for both sexes.

Blood lipid levels are frequently found to be disturbed in childhood cancer diagnosis [68,69] and tend to improve after treatment [69,70]. Children with hematologic cancers and/or treated with L-asparaginase, corticosteroids, or cranial irradiation are more likely to develop dyslipidemia [71]. Here, we found an increase in the mean HDL-C levels between the two assessments, but the sample size did not allow for sub-analyses according to the cancer type or treatment. Such an improvement in the HDL-C was not detected elsewhere after a healthy lifestyle intervention study, nor were the levels different to that of a control group not exposed to the intervention [61]. Some critical differences can explain the outcome disparities between this study and ours including a shorter intervention duration (4 vs. 12 months), being overweight as an inclusion criterion (vs. not), and the period of the beginning of the intervention (remission vs. early after diagnosis). As a matter of fact, overweight people tend to have lower HDL-C [72]. Additionally, since the lipid profile tends to return to normal after the treatments [70,73], variations could be less important if the intervention is implemented during remission. Notably, despite this reported normalization of lipid levels, CCS still suffer disproportionally from lipid abnormalities compared to the general population [24,26]. Given that dietary intakes were associated with HDL-C levels in a cohort of ALL survivors [74], it is possible to hypothesize that a better diet during antineoplastic therapy has beneficial effects on HDL-C levels in young patients. In our study, levels of 25-hydroxy vitamin D also increased after the one-year intervention. However, the intake of vitamin D or multivitamin supplements were not systemically recorded during the intervention, apart from the data collected during the RD visits. In contrast, no improvement in 25-hydroxy vitamin D status was found in a 4-patient nutrition intervention feasibility study [63]. Serum vitamin D levels are commonly insufficient at pediatric cancer diagnosis [75,76] and during treatment [77]. A retrospective study further demonstrated that vitamin D levels in this population decrease significantly over time during treatment [78]. Despite the encouraging results of our study, more research is needed to investigate whether nutrition interventions during pediatric cancer treatment can improve nutritional status.

Our study had several strengths. First, compared to other nutrition intervention studies in pediatric oncology, the cohort size was relatively large. We identified only two similar studies with larger cohorts in the literature: the SHARE study [79] including 38 patients and consisting of a half-day workshop intervention where assessments were made at the baseline and 1 month after, and a multicenter study including 132 adolescents in remission to whom lifestyle multi-component counselling intervention and phone follow-ups were offered for one year [59]. Second, our nutrition intervention was deployed over a one-year period, which is superior to most studies, and was individualized to the needs of each patient and family. Third, using diet scores validated for the pediatric population allowed us to monitor the evolution of diet quality. Most importantly, our study’s uniqueness resides in the fact that the intervention was initiated early after pediatric cancer diagnosis (<12 weeks).

The study limitations include the unknown impact of the COVID-19 pandemic on the outcomes. While some families reported little differences in dietary habits, others shared that healthy eating was no longer a priority during the pandemic or that, conversely, some cooked more at home during this period. Additionally, for feasibility reasons, food intakes were gathered using 24-h dietary recalls, although it has been established that food frequency questionnaires and 3-day food diaries are more comprehensive methods of collecting dietary data [80,81]. A main limitation of this study is the small sample size, although the number of participants was greater or comparable to those found in other monocentric studies in pediatric oncology proposing similar interventions [34,58,60,63,64,65]. Importantly, the absence of a control group prevents us from interpreting changes as uniquely attributable to the intervention. Moreover, from the VIE study, being a multiple component project including physical activity and psychological support, it is plausible that some outcomes are not solely attributable to the nutrition intervention. Furthermore, the observed changes in diet, anthropometric, and biochemical parameters could have been influenced by many factors other than the intervention such as cancer evolution, the side effects of treatment, and exposure to medications (e.g., corticosteroids). Studying the impact of these confounding variables on the study outcomes was not feasible given the limited sample size and population heterogeneity. Finally, given the limited sample size and other limitations, it is premature to generalize our findings to all children treated for cancer. More studies are needed to confirm our results.

## 5. Conclusions

This study supports that a one-year nutritional intervention deployed early after pediatric cancer diagnosis was associated with an improvement in the diet of children and adolescents. This is accompanied by changes in the biophysical profiles as well as lipid profile (HDL-C) and nutritional status (25-hydroxy vitamin D). While the study design did not allow us to confirm that these changes were solely attributable to the intervention, the results support the importance of offering tailored nutritional strategies early in the cancer journey. More research is needed to assess their effectiveness and impact on diet and health during and after pediatric cancer.

## Figures and Tables

**Figure 1 children-10-00667-f001:**
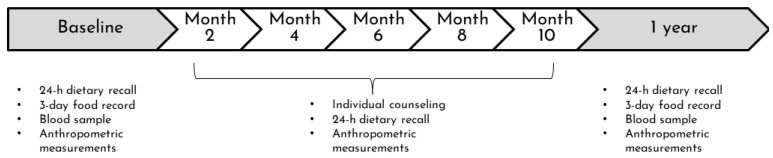
Design of the nutritional intervention of the VIE project. The intervention consisted of an initial assessment, follow-up visits planned every two months for one year, and a one-year assessment.

**Figure 2 children-10-00667-f002:**
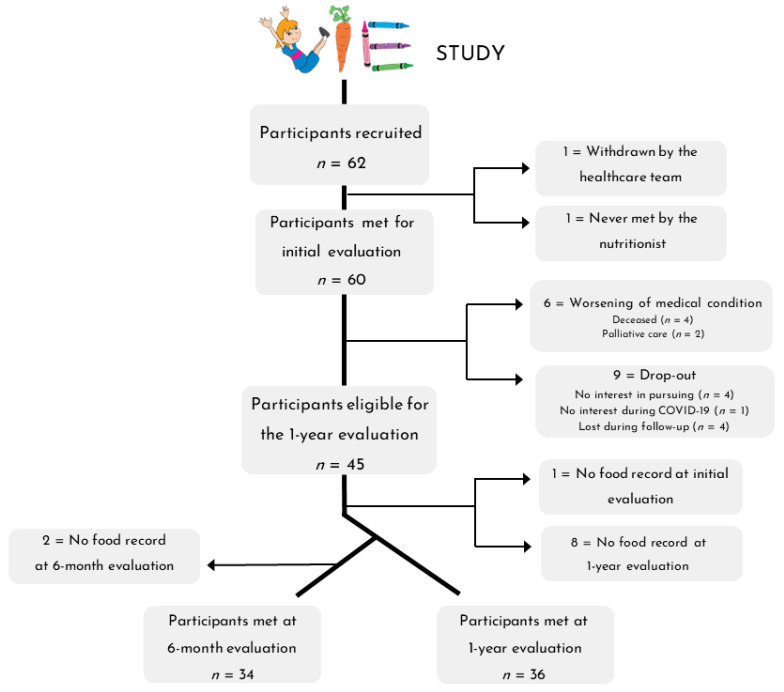
Flow diagram of patient recruitment for the nutritional intervention.

**Figure 3 children-10-00667-f003:**
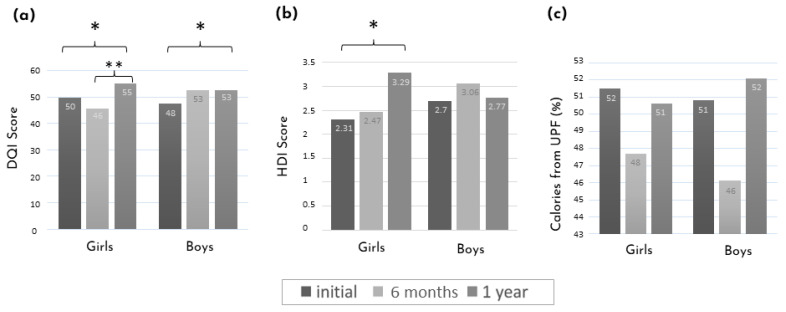
Mean diet score evolution between the initial assessment, after 6 months, and one year of intervention stratified by sex. (**a**) DQI score. (**b**) HDI score. (**c**) Calories from UPF (%). * *p*-value < 0.05 and ** *p*-value < 0.01 using the paired t-tests or Wilcoxon between time points. DQI: diet quality index; HDI: healthy diet index; UPF: ultra-processed foods.

**Table 1 children-10-00667-t001:** Characteristics of the participants.

Characteristics	Participants
	*n* = 36
Sex, *n* (%)	
Male	19 (52.8)
Female	17 (47.4)
Age, years	
Mean ± SD Min–max	7.88 ± 5.00 1.57–17.1
Diagnosis, *n* (%)	
Leukemia ^1^	18 (50.0)
Lymphoma ^2^	9 (25.0)
Sarcoma ^3^	4 (11.1)
Others ^4^	5 (13.9)
Number of visits	
Mean ± SD Min–max	4.72 ± 1.06 2.00–6.00
4 visits and less, *n* (%)	12 (33.3)
5 visits and more, *n* (%)	24 (66.7)
Level of engagement	
Low and moderate, *n* (%)	13 (36.1)
High, *n* (%)	23 (63.9)

^1^ Leukemia diagnosis includes acute lymphoblastic leukemia (*n* = 16), acute myeloid leukemia (*n* = 2). ^2^ Lymphoma diagnosis includes Hodgkin’s lymphoma (*n* = 2), Burkitt’s lymphoma (*n* = 5), anaplastic lymphoma (*n* = 1), and lymphoblastic lymphoma (*n* = 1). ^3^ Sarcoma diagnosis includes rhabdomyosarcoma (*n* = 3), Ewing’s sarcoma (*n* = 1). ^4^ Other diagnoses include Wilm’s tumor (*n* = 1), germinoma (*n* = 1), medulloblastoma (*n* = 1), neuroblastoma (*n* = 1), and hepatoblastoma (*n* = 1). Min: minimum; Max: maximum; SD: standard deviation.

**Table 2 children-10-00667-t002:** Dietary intakes and scores of diet quality at the initial assessment, after 6 months, and one year of nutritional intervention.

Characteristics of Diet	Initial Assessment	After 6 Months of Nutritional Follow-Up	After One Year of Nutritional Follow-Up	Difference (6 Months–Initial)	*p*-Value	Difference (One Year–6 Months)	*p*-Value	Difference (One Year–Initial)	*p*-Value
	*n* = 36	*n* = 34	*n* = 36	*n* = 34		*n* = 34		*n* = 36	
	Mean ± SD (min–max)	Mean ± SD (min–max)	Mean ± SD (min–max)	Mean ± SD		Mean ± SD		Mean ± SD	
Diet Quality Scores									
Diet Quality Index	48.53 ± 10.4 (12.0–66.0)	49.51 ± 13.8 (4–74.5)	53.75 ± 9.67 (38.0–74.5)	0.57 ± 13.4	0.805	4.75 ± 12.98	0.093	5.22 ± 9.95	0.003
Healthy Diet Index	2.51 ± 1.02 (0–4.25)	2.80 ± 1.36 (0.67–7)	3.02 ± 1.44 (1–7)	0.23 ± 1.48	0.613	0.16 ± 1.82	0.613	0.05 ± 1.58	0.065
Low adherence, *n* (%)	31 (86.1)	22 (64.71)	22 (61.11)		0.027		0.209		0.012
Moderate and strong adherence, *n* (%)	5 (13.9)	12 (35.29)	14 (38.9)						
NOVA (%)	51.1 ± 22.5 (0–100)	46.8 ± 22.96 (7.24–100)	51.4 ± 20.6 (12.0–98.0)	−0.04 ± 0.25	0.372	0.03 ± 0.30	0.561	0.002 ± 0.28	0.96
Nutrients									
Energy (kcal/d)	1538 ± 742 (374–4285)	1691 ± 766 (5466–3606)	1871 ± 752 (925–4333)	176 ± 975	0.300	234 ± 622	0.035	333 ± 1030	0.033
Total fat (g/1000 kcal/d)	38.3 ± 8.12 (21.7–55.3)	38.3 ± 11.5 (13.4–59.3)	37.5 ± 9.24 (23.7–57.9)	0.65 ± 15.3	0.807	−0.59 ± 15.4	0.826	−0.80 ± 11.5	0.68
Protein (g/1000 kcal/d)	40.9 ± 8.55 (12.6–56.2)	46.8 ± 13.5 (26.5–84.2)	40.9 ± 10.0 (19.7–68.5)	5.51 ± 14.9	0.072	−5.89 ± 17.4	0.057	−0.08 ± 13.2	0.97
Carbohydrate (g/1000 kcal/d)	125.91 ± 19.15 (82.1–164.7)	118.9–24.8 (65.96–165.7)	128 ± 23.03 (78.97–171.5)	−8.22 ± 32.7	0.152	8.68 ± 28.4	0.085	2.10 ± 25.7	0.63
Omega-3 polyunsaturated fatty acids (g/1000 kcal/d)	0.33 ± 0.39 (0–1.77)	0.44 ± 0.47 (0–1.77)	0.74 ± 2.40 (0–14.5)	0.11 ± 0.58	0.283	0.34 ± 2.48	0.431	0.41 ± 2.46	0.33
Fiber (g/1000 kcal/d)	8.09 ± 2.76 (0–14.6)	8.03 ± 3.33 (2.85–16.2)	7.82 ± 2.73 (1.67–15.1)	−0.26 ± 4.33	0.732	−0.17 ± 3.98	0.904	−0.27 ± 4.26	0.71

Dietary intakes were collected using 24-h dietary recall and/or a 3-day food record. Data were analyzed with Nutrific software. Paired t-tests or Wilcoxon tests and McNemar’s tests were used to compare the diets between initial assessment and after 6 months and one year of nutritional intervention. *p*-value < 0.05 is considered statistically significant; SD: standard deviation. NOVA: % represents the proportion of calories provided by ultra-processed foods.

**Table 3 children-10-00667-t003:** Clinical and anthropometric characteristics of the participants at the initial assessment and after one year of nutritional intervention.

Characteristics		Initial Assessment		After One Year of Nutritional Follow-Up		Difference (One Year–Initial)	*p*-Value
	*n* (total)	Mean z-score ± SD (min-max)	*n* (total)	Mean z-score ± SD (Min-max)	*n* (pairs)	Mean z-score (SD)	
Weight	36	0.14 ± 1.13 (−2.16–3.03)	36	0.43 ± 1.18 (−1.94–3.37)	36	0.29 (0.70)	0.019
Height	36	−0.21 ± 1.08 (−2.47–3.03)	36	−0.39 ± 1.03 (−3.13–1.59)	36	−0.17 (0.55)	0.029
BMI	36	0.37 ± 1.11 (−1.64–2.81)	36	0.87 ± 1.03 (−0.69–3.09)	36	0.50 (0.88)	0.002
Waist circumference	12	0.87 ± 0.63 (0.13–1.84)	12	0.74 ± 1.17 (−0.84–2.13)	5	−0.21 (0.72)	0.547
MUAC	23	0.37 ± 1.30 (−2.01–2.37)	19	0.80 ± 1.00 (−0.70–2.51)	13	0.23 (1.22)	0.51
TSFT	18	0.20 ± 1.18 (−2.75–2.08)	12	0.66 ± 0.70 (−0.50–1.96)	10	0.75 (1.29)	0.10
SSFT	15	0.18 ± 1.20 (−2.50–1.95)	9	0.06 ± 0.91 (−1.39–1.28)	7	0.32 (1.25)	0.52
Systolic blood pressure	36	0.37 ± 0.99 (−1.89–2.14)	32	0.38 ± 0.85 (−1.49–2.16)	32	−0.003 (1.18)	0.99
Diastolic blood pressure	35	0.37 ± 0.91 (−1.63–2.17)	32	0.31 ± 1.11 (−1.26–2.88)	31	−0.17 (0.83)	0.32

Anthropometric and clinical data were collected at the initial visit and after one year of nutritional intervention. Data at the initial assessment and after one year were compared using paired *t*-tests or Wilcoxon tests. *p*-value < 0.05 is considered statistically significant. BMI: body mass index [weight (kg)/height (m^2^)]; MUAC: mid-upper arm circumference; TSFT: triceps skinfold thickness; SSFT: subscapular skinfold thickness; SD: standard deviation.

**Table 4 children-10-00667-t004:** Biochemical data of the participants at the initial assessment and after one year of nutritional intervention.

Biochemical Data		Initial Assessment		After One Year of Nutritional Follow-Up		Difference (One Year–Initial)	*p*-Value
	*n* (total)	Mean ± SD (Min-max)	*n* (total)	Mean ± SD (Min-max)	*n* (pairs)	Mean (SD)	
HbA1c (%)	33	5.12 ± 0.56 (3.60–6.00)	24	4.98 ± 0.43 (4.10–5.80)	22	−0.18 (0.82)	0.32
Vitamin D (mmol/L)	31	54.2 ± 15.6 (25.3–85.3)	24	68.99 ± 27.44 (28.3–160.5)	21	14.52 (28.1)	0.03
C-reactive protein (mmol/L)	32	7.44 ± 16.5 (0.20–85.6)	24	4.09 ± 7.24 (0.20–29.30)	22	−4.41 (21.4)	0.35
Total cholesterol (mmol/L)	35	4.08 ± 2.03 (1.69–11.9)	25	4.01 ± 1.04 (2.98–8.10)	25	−0.20 (2.04)	0.63
HDL-C (mmol/L)	34	0.94 ± 0.29 (0.31–1.55)	25	1.22 ± 0.29 (0.66–1.90)	24	0.27 (0.37)	0.002
LDL-C (mmol/L)	34	2.62 ± 1.89 (0.89–9.83)	25	2.28 ± 1.07 (0.75–6.47)	24	−0.48 (1.85)	0.22
Non HDL-C (mmol/L)	34	3.18 ± 2.06 (1.27–11.16)	25	2.79 ± 1.08 (1.77–6.98)	24	−0.51 (2.10)	0.25
Triglycerides (mmol/L)	35	1.31 ± 0.84 (0.30–2.94)	25	1.11 ± 0.75 (0.48–3.71)	25	−0.19 (0.89)	0.30
0–9 years (mmol/L)	23	1.29 ± 0.86 (0.45–2.94)	16	0.82 ± 0.37 (0.48–1.90)	16	−0.43 (0.81)	0.021
10–18 years (mmol/L)	12	1.33 ± 0.84 (0.30–2.85)	9	1.63 ± 0.98 (0.58–3.71)	18	0.25 (1.00)	0.510

Blood samples were collected during clinical visits. Data at the initial assessment and after one year of nutritional follow-up were compared using paired *t*-tests or Wilcoxon tests. *p*-value < 0.05 was considered statistically significant. HbA1C: glycosylated hemoglobin; HDL-C: high-density lipoprotein-cholesterol; LDL-C: low-density lipoprotein cholesterol; Non HDL-C: non high-density lipoprotein-cholesterol; SD: standard deviation.

## Data Availability

Not applicable.

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
