# Peer review of "Improvement of Diet after an Early Nutritional Intervention in Pediatric Oncology"

_children, 2023, doi:10.3390/children10040667_

Round 1

Reviewer 1 Report

This study by Mélanie Napartuk supports that a one-year nutritional intervention deployed  early after a pediatric cancer diagnosis is associated with an improvement of children's and adolescents’ diet. Their work provide practical suggestions for nutritional assessment in pediatric cancer patients.

Author Response

We thank the Reviewer for his/her positive comments and encouragements. Different precisions were made regarding the methods, which we believe enhance the quality of our manuscript.

Reviewer 2 Report

Interesting scientific article.

The paper does not report whether anthropometric measurements are standardized using the ISAK model. Were all the operators trained to measure the anthropometric measurements in the same areas? Anthropometry presents inter-operator and intra-operator variability. Was the waist circumference measured with reference ISAK, WHO, or NIH?

I suggest making the following corrections to the paper text:

Line 45: complications (11-13). = Remove comma (,) afterword complications

Line 79: documented (46). = Add a space afterword documented

Line 150: 0.1 cm = Add a space afterword 0.1

Table 2: p-value = value with "v" in lowercase

Line 230/257/272: p-value = p in lowercase

Reviewer 3 Report

The manuscript under the title "Improvement of diet after an early nutritional intervention in pediatric oncology" deals with an interesting topic, namely nutrition quality of children diagnosed with cancer. There are several methodological issues that need to be addressed before it is considered suitable for publication. 

1. The number of children recruited is too low and the population is children and adolescents. How many children and how many adolescents? 

2. Did you do power calculation for your results? I didn't see a normality test. Please provide data of power and normality and amend your statistical analysis accordingly as now it is not correct. 

3. How did you do the recruitment of the children? Please provide relative data

4. As you mention in the limitations you do not have a control group and having in mind the small sample size this limit even more the validity of your data

5. Do you have data regarding the type of treatment and any restrictions in nutrition for the children? How many were on corticosteroids or other medication that could impact glucose metabolism, appetite, and weight?

6. You present data on cholesterol and glucose metabolism. Why did you choose these parameters and not others related to nutritional status/ malnutrition?

7. The limitation paragraph should be expanded. COVID-19 is of minor importance compared to the other problems of your methodology. 

8. 2.4 Dietary data: Change into Dietary assessment data. 

9. In the table presenting the results of anthropometry several children were not measured. Please provide reasonable justification. I would suggest that you omitted the variables not including the total sample as your sample is already too small. 

Reviewer 4 Report

This is a great original , well designed work.

Author Response

We thank the Reviewer for his/her positive comments and encouragements.

Round 2

Reviewer 3 Report

Thank you for your attempt to answer some of the comments. The main concern remains, i.e. the underpower of your study and methodological errors that lower the quality of the presented work. Acknowledging the limitations of your study but resolving the problems is required.. Therefore, I do believe that you need to recruit enough patients and then seek a journal for publication